# Diagnostic accuracy of history taking, physical examination and imaging for non-chronic finger, hand and wrist ligament and tendon injuries: a systematic review update

Patrick Krastman  ,[1] Nina M C Mathijssen,[2] Sita M A Bierma-Zeinstra,[1,3] Gerald A Kraan,[2] Jos Runhaar[1]

[1]Department of General Practice, Erasmus MC University Medical Center, Rotterdam, The Netherlands
[2]Department of Orthopaedic, Reinier de Graaf Gasthuis, Delft, The Netherlands
[3]Department of Orthopedics, Erasmus MC University Medical Center, Rotterdam, The Netherlands

**Correspondence to**
Patrick Krastman;
wetenschap@dezorghoek.nl

## ABSTRACT

**Objective** The diagnostic work-up for ligament and tendon injuries of the finger, hand and wrist consists of history taking, physical examination and imaging if needed, but the supporting evidence is limited. The main purpose of this study was to systematically update the literature for studies on the diagnostic accuracy of tests for detecting non-chronic ligament and tendon injuries of the finger, hand and wrist.

**Methods** Medline, Embase, Cochrane Library, Web of Science, Google Scholar ProQuest and Cinahl were searched from 2000 up to 6 February 2019 for identifying studies. Methodological quality was assessed using the Quality Assessment of Diagnostic Accuracy Studies 2 checklist, and sensitivity (Se), specificity (Sp), accuracy, positive predictive value (PPV) and negative predictive value (NPV) were extracted.

**Results** None of the studies involved history taking. Physical examination, for diagnosing lesions of the triangular fibrocartilage complex (TFCC), showed Se, Sp, accuracy, PPV and NPV ranging from 58% to 90%, 20% to 69%, 56% to 73%, 53% to 71% and 55% to 65%, respectively. Physical examination in hand and finger injuries the Se, Sp, accuracy, PPV and NPV ranged from 88% to 99%, 75% to 100%, 34% to 88%, 91% to 100% and 75% to 95%, respectively. The accuracy of MRI with high-resolution (3 T) techniques for TFCC and interosseous ligaments of the proximal carpal row ranged from 89% to 91% and 75% to 100%, respectively. The accuracy of MRI with low-resolution (1.5 T) techniques for TFCC and interosseous ligaments of the proximal carpal row ranged from 81% to 100% and 67% to 95%, respectively.

**Conclusions** There is limited evidence on the diagnostic accuracy of history taking and physical examination for non-chronic finger, hand and wrist ligament and tendon injuries. Although some imaging modalities seemed to be acceptable for the diagnosis of ligament and tendon injuries in the wrist in patients presenting to secondary care, there is no evidence-based advise possible for the diagnosis of non-chronic finger, hand or wrist ligament and tendon injuries in primary care.

### Strengths and limitations of this study

► This is the first study that systematically reviewed the accuracy of diagnostic tests for non-chronic hand and finger injuries, next to previously described accuracy of diagnostic tests for non-chronic wrist injuries.

► Studies on wrist injuries published before 2000 were not evaluated and not included in the current systematic review, as these were adequately described in published systematic reviews.

► Diagnostic tests heterogeneity precluded meta-analysis, caused by the fact that studies that evaluated the same pathologies showed marked diversity in population, index tests, reference test and methodological quality.

## INTRODUCTION

Wrist injuries are one of the most common presentations to the emergency department (ED) due to trauma and they commonly affect young people of working age.[1 2] In the Netherlands, 21% of the patients initially consulted their general practitioner (GP) after a wrist injury, 41% went directly to an outpatient clinic and 35% had no further treatment.[3] Within the GP's practice, the prevalence of hand injuries is 10 for each 1000 patients per year, while the prevalence for wrist injuries is 6 for each 1000 patients per year.[4] In an ED, injuries to the hand and wrist are common and they account for between 10% and 30% of all presentations.[3 5–7] Traumatic hand injuries are a frequent part among work-related injuries and can result in prolonged sick leave. They represent a considerable economic burden, with both high healthcare and productivity costs.[5] If not treated properly, patients may experience lifelong pain and functional limitations that have major

effects on the quality of life and could result in patients losing their jobs.[8]

The standard diagnostic work-up for non-chronic finger, hand and wrist trauma consists of history taking, a physical examination and, if needed, imaging. There is general agreement that a detailed patient history and a conscientious clinical examination should be standard methods of diagnosing wrist pain.[9] Nevertheless, the diagnosis of wrist pathologies remains complex and challenging, since the wrist contains many joints that function together to move the hand, and there is increasing demand for evidence for diagnostic technologies, such as imaging tools.[10]

Evidence-based medicine is required to create well-founded policies for non-chronic finger, hand and wrist ligament and tendon injuries. It is essential to distinguish between diagnosing these injuries in hospital care and in non-institutionalised GP care, as results from diagnostic studies in hospital care cannot automatically be translated into guidelines for non-institutionalised GP care.[11] Diagnostic accuracy is affected by the prevalence of the pathology. Predictive values are largely dependent on the prevalence of the pathology in the examined population. Therefore, predictive values from one study should not be transferred to another setting with a different prevalence of the disease in the population.[12] Nevertheless, currently available systematic reviews on the diagnostic accuracy of tests for the diagnosis of finger, hand and wrist pathologies did not distinguish between hospital and non-institutionalised GP care settings when presenting their results.[10 13–15] Within the available systematic reviews, published up to 2015, no studies were found on the diagnostic accuracy of history taking and only the scaphoid shift test and high-resolution MRI were recommended for diagnosing triangular fibrocartilage complex (TFCC) tears.[10 13–15]

The main purpose of the present study was to provide a systematic overview of the diagnostic accuracy of history taking, physical examination and imaging for detecting non-chronic ligament and tendon injuries of the finger, hand and wrist. The secondary aim of this study was to retrieve the clinical care setting (hospital or non-institutionalised GP) of the eligible studies and the studies published in previous systematic reviews.

## METHODS

The Preferred Reporting Items for Systematic Reviews and Meta-Analyses statement was used to guide the conduct and reporting of the study.[16] A review protocol was composed prior to searching the literature, but central registration was not completed.

### Search strategy

A biomedical information specialist (Wichor M Bramer) from the Medical Library at Erasmus MC performed a search for studies in Medline, Embase, Cochrane Library, Web of Science, Google Scholar ProQuest and Cinahl

from 2000 up to 6 February 2019. This starting point was used since multiple reviews are available that already cover the period up to the year 2000 (table 1). Search terms included hand, finger and wrist injuries, history taking, provocative test(s), diagnostic test(s) and imaging tests. The full electronic search strategy for the Embase database is presented in online supplemental appendix 1.

### Study selection criteria

Studies describing diagnostic accuracy of history taking, physical examination or imaging in adult patients (age ≥16 years) with non-chronic finger, hand and wrist ligament and tendon injuries were included. Diagnostic accuracy was rabeported or could be calculated. Case reports, reviews and conference proceedings were excluded. Distal radius and ulna injuries were also excluded. Chronic injuries (eg, osteoarthritis) were excluded as a result of another pathophysiology. There was no gold-standard reference test against which to assess history taking, physical examination or imaging measurements. Surgical observations (arthroscopy) are the reference standards for confirming a diagnosis of non-chronic hand, finger or wrist injury, although only a subset of patients suspected of having non-chronic hand, finger or wrist injury require surgery. To decrease verification bias, diagnostic-imaging techniques for non-chronic hand injury were accepted as reference tests as well. Since tendinopathy does not typically require surgery, imaging is also a pragmatic reference standard for this condition. As this review focused on non-chronic pathologies, studies, including patient with chronic pathologies (eg, osteoarthritis and rheumatic arthritis), were excluded. Infection and neurological injuries are out of the scope of this review and are, therefore, not included. Carpal tunnel syndrome is extensively described in the literature and was, therefore, not included in this review.[17–19] Diagnoses of musculoskeletal soft-tissue tumours were also excluded. No language restrictions were applied. For languages of the eligible studies other than English, Google translate was used for the first translation of these studies. If necessary, a professional translator was consulted.[20]

Two reviewers (PK and Yassine Aaboubout) read all titles and abstracts independently. Articles that could not be excluded on the basis of the title and/or abstract were retrieved in full text and were read and checked for inclusion by the two reviewers independently. If there was no agreement, a third reviewer (JR) made the final decision. In addition, the reference lists of all included studies were reviewed to check for additional relevant studies.

### Data collection process and methodological quality assessment

In the current review, our primary outcome measures were the positive predictive value (PPV) and the negative predictive value (NPV) of diagnostic tests. Secondary outcome measure were the sensitivity (Se), specificity (Sp) and accuracy of diagnostic tests.

**Table 1** Characteristics of the eligible studies (N=23)

| Author (year) | Participants | Design | Setting (country) | Trauma | Index test 1 | Index test 2 | Reference test |
|---|---|---|---|---|---|---|---|
| Wrist injuries | | | | | | | |
| Anderson et al (2008)[23] | 102 | Retrospective | Not described (USA) | TFCC/SLIL/LTIL/UTIL | MRI (1.5 T) | MRI (3 T) | Arthroscopy |
| Pahwa et al (2014)[32] | 53 | Prospective | Not described (India) | TFCC/SLIL/LTIL | MRI (1.5 T) | MR arthrography | Arthroscopy |
| Prosser et al (2011)[33] | 105 | Prospective | Private hand clinic (Australia) | TFCC/SLIL/LTIL | MRI (1 T) | Provocative tests | Arthroscopy |
| Langner et al (2015)[40] | 38 | Not described | Not described (Germany) | SL dissociation | Cine MRI (3 T) | Cineradiography | Arthroscopy |
| Spaans et al (2013)[41] | 37 | Not described | Department for hand and plastic surgery* (The Netherlands) | SLIL (complete tear) | MRI (3 T) | | Arthrotomy |
| Greditzer et al (2016)[24] | 26 | Retrospective | Department for hand and plastic surgery* (USA) | SLIL | MRI (1.5 T) axial sequences | MRI (1.5 T) coronal sequences | Arthroscopy |
| Al-Hiari (2013)[34] | 42 | Prospective | Orthopaedic surgery* (Jordan) | TFCC (full-thickness tears) | MR arthrography | | Arthroscopy |
| Schmauss et al (2016)[25] | 908 | Retrospective | Department for hand and plastic surgery (Germany) | TFCC | MRI (resolution not described) | Provocative tests | Arthroscopy |
| Lee et al (2016)[35] | 39 | Prospective | Not described (China) | TFCC (full-thickness tears)/SLIL/LTIL | MR (3 T) arthrography without traction | MR (3 T) arthrography with traction | Conventional arthrography |
| Finlay et al (2004)[26] | 26 | Retrospective | Not described (Canada) | TFCC/SLIL/LTIL | US (9–13 MHz) | | MR arthrography† |
| Dornbergeret al (2015)[36] | 72 | Prospective | Hand surgery* (Germany) | SLIL | Radiographs | | Arthroscopy |
| Koskinen et al (2012)[42] | 52 | Not described | Not described (Finland) | TFCC/SLIL/LTIL | CBCT arthrography | | MR arthrography |
| Boer et al (2018)[27] | 150 | Retrospective | Plastic or orthopaedic surgery (The Netherlands) | TFCC | MRI (1.5 T or 3.0 T) | MR arthrography (1.5 or 3.0 T) | Arthroscopy |
| Lee and Yun (2018)[31] | 65 | Prospective | ED (Korea) | TFCC | US | | MRI (3.0 T) |
| Suojärvi et al (2017)[37] | 21 | Prospective | Hand surgery (Finland) | SLIL/LTIL/TFCC | CBCT arthrography | MR arthrography | Arthroscopy |
| Mahmood et al (2012)[30] | 30 | Retrospective | General hospital (UK) | SLIL/LTIL/TFCC | MR arthrography | | Arthroscopy |
| Hand and finger injuries | | | | | | | |
| Lutsky et al (2014)[28] | 20 | Retrospective | Not described (USA) | Collateral ligament tears of the MPJ of the fingers | MRI (open,1.5 T and 3 T) | | Surgical findings |
| Guntern et al (2007)[29] | 8 | Retrospective | Not described (Switzerland) | A2 pulley lesion | Clinical examination | | MRI (3 T) |
| Klauser et al (2002)[43] | 64 | Not described | Not described (Austria) | Finger pulley injuries | US (12 MHz) | | MRI (1.5 T) |
| Lee et al (2000)[44] | 10 | Not described | Not described (USA) | Flexor tendon injuries | US (L10–5 MHz) | | Surgical findings |

Continued

**Table 1** Continued

| Author (year) | Participants | Design | Setting (country) | Trauma | Index test 1 | Index test 2 | Reference test |
|---|---|---|---|---|---|---|---|
| Zhang et al (2012)[45] | 92 | Not described | Department of surgery (China) | Flexor tendon injuries | US (10 MHz) | | Surgical findings |
| Mahajan et al (2016)[39] | 30 | Prospective | Emergency room and outpatients clinic of surgery and orthopaedics (the Netherlands) | UCL injuries | Clinical examination | | MRI (1.5 T) |
| Shekarchi et al (2017)[38] | 20 | Prospective | ED (Iran) | UCL of the thumb | US | | MRI |

*Setting for the study was obtained after email contact.
†Tricompartment wrist arthrography.
CBCT, cone-beam CT; ED, emergency department; LTIL, lunotriquetral interosseous ligament; MPJ, metacarpophalangeal joint; MR, magentic resonance; SLIL, scapholunate interosseous ligament; TFCC, triangular fibrocartilage complex; UCL, ulnar collateral ligament; US, ultrasonography; UTIL, ulnotriquetral interosseous ligament.

Two reviewers (PK and JR) independently extracted the data. Data were extracted describing the study design, characteristics of the study population, test characteristics, setting (hospital care or non-institutionalised GP care) and diagnostic parameters. The following values were extracted, when documented: Se, Sp, accuracy, PPV and NPV. If diagnostic parameters were not reported, they were calculated from reported data or authors were contacted by email when data were unavailable. The following formula was used, when calculating diagnostic accuracy: diagnostic accuracy=(the number of true positives+the number of true negatives)/total number of subjects.[21] If an included study presented results from multiple independent observers, accuracy measures were averaged over the observers. Furthermore, data of the studies published in previous systematic reviews were extracted describing the setting (hospital care or non-institutionalised GP care). Methodological quality was assessed using the Quality Assessment of Diagnostic Accuracy Studies 2 (QUADAS-2) checklist.[22] This tool allows more transparent rating of bias and applicability in primary diagnostic accuracy studies. The QUADAS-2 tool consists of four domains: patient selection, index test, reference standard, and flow and timing. Two reviewers (PK and JR) independently assessed the risk of bias and applicability of each included study. Disagreements were resolved by discussion. Questions were answered with 'yes', 'no' or 'unclear'.

### Patient and public involvement
Patients and members of the public were not involved in this systematic review update.

### RESULTS
### Study selection
The flow diagram for the categorisation process is presented in figure 1. We assessed 209 full-text articles for eligibility out of 4867 records identified through database searches. A total of 23 diagnostic studies were finally identified, assessed and interpreted.

### Study characteristics
The characteristics of the studies are presented in table 1.
Eight studies were retrospective[23–30] nine studies were prospective[31–39] and six studies[40–45] gave no description of the study design. Eight studies[23 25 27 31 33 36 43 45] included more than 60 participants; six of these studies[23 25 27 31 33 36] described wrist pathologies and two[43 45] described hand pathologies. In total, 16 studies[23–27 30–37 40–42] described injuries to the wrist anatomy and seven studies[28 29 38 39 43–45] described injuries to the hand/finger anatomy.

### Quality assessment
There was considerable underreporting of important quality domains in most studies (see table 2).
Two studies had low risk of bias on all quality domains.[31 33] In 8[24 26 27 30 32 38 41 44] of the 23 studies, patient selection

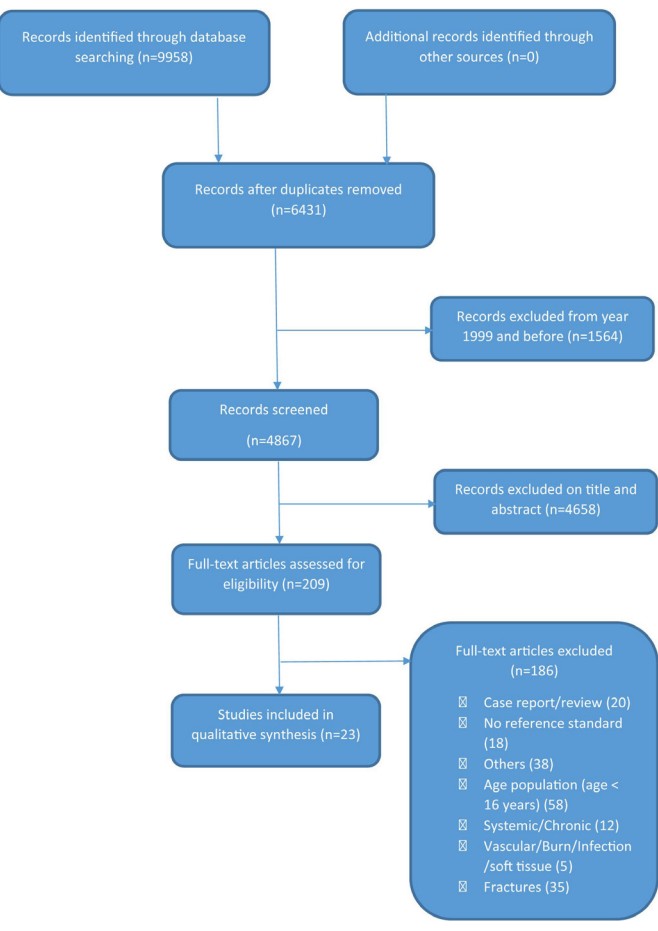

**Figure 1** Flow chart study selection.

was not well documented. Furthermore, the risk of bias was predominantly influenced by the lack of a proper description of the index test (30%, 7/23)[25 27–29 35 42 43] or the reference standard (65%, 15/23).[23–25 27 28 30 32–37 40 42 45] Regarding flow and timing, not all patients received the reference standard in four studies (22%, 5/23).[32 34 37 41 43] Due to our selection procedure, all the studies match the review question.

### Accuracy of diagnostic tests concerning wrist injuries

None of the studies evaluated the diagnostic accuracy of history taking. Physical examination was evaluated in two studies for diagnosing lesions of the TFCC.[25 33] Provocative wrist tests for diagnosing scapholunate interosseous ligament (SLIL) and lunotriquetral interosseous ligament (LTIL) lesions was assessed in one study.[33]

Radiographs were used as an index test in one study for diagnosing SLIL lesions.[36] Ultrasonography (US) for diagnosing TFCC lesions was used in two studies.[26 31] Two studies used cone-beam CT (CBCT) as index for diagnosing TFCC lesions.[37 42] In 12 studies, MRI was used as an index test.[23–25 27 30 32–35 37 40 41] The accuracy of MRI for TFCC, SLIL, LTIL and ulnotriquetral interosseous ligament (UTIL) lesions with high-resolution (3 T) techniques ranging from 89% to 91%, 75% to 92%, 91% and 100%, respectively. The accuracy of MRI for TFCC, SLIL, LTIL and UTIL lesions with low-resolution (1.5 T)

techniques ranging from 81% to 100%, 67% to 81%, 81% to 94% and 95%, respectively. The accuracy measures of the diagnostic tests are presented in table 3.

In addition to the data presented in table 3, the study of Schmauss *et al* presented the diagnostic accuracy of their tests separately for different subgroups.[25] These results are summarised in online supplemental appendix 2.

### Accuracy of diagnostic tests concerning hand and finger injuries

Table 4 describes the accuracy of the diagnostic tests for non-chronic hand and finger injuries.[28 29 38 39 43–45]

Two studies concerned flexor tendon injuries,[44 45] while the other studies concerned collateral ligament tears of the metacarpophalangeal joint of the fingers,[28] A2 pulley lesions,[29] finger pulley injuries[43] and ulnar collateral ligament (UCL) injuries.[38 39] None of the studies involved history taking. Clinical examination was used three times as an index test.[29 39 45] The Se, Sp, accuracy, PPV and NPV of physical examination in hand and finger injuries ranged from 88% to 99%, 75% to 100%, 34% to 88%, 91% to 100% and 75% to 95%, respectively. MRI was used once as an index test.[28] Four studies used ultrasonography (US) as an index test.[38 43–45] The accuracy of US in flexor injuries ranged from 90% to 100%.[44 45] The accuracy of US for finger pulley injuries and UCL of the thumb was 99% and 80%, respectively.[38 43]

### Clinical care setting

The clinical care setting was described in 9 out of 23 studies and was obtained by contacting the authors for an additional 4 studies: a private hand clinic,[33] ED,[31 38] department for hand and plastic surgery,[24 27 34 36 41] surgery,[45] orthopaedics department[27 34] and in an emergency room and outpatient clinic of a surgery and orthopaedics department.[39] Despite multiple attempts to contact the authors by email, clarification regarding the setting could not be obtained for the remaining 10 studies.

### DISCUSSION

The standard diagnostic work-up for non-chronic finger, hand and wrist trauma consists of history taking, a physical examination and, if needed, imaging. There is general agreement that a detailed patient history and a conscientious clinical examination should be standard methods of diagnosing wrist pain.[9] Our systematic review showed that there is still a gap in knowledge regarding valid diagnostic tests for non-chronic wrist ligament and tendon injuries. Moreover, for the first time, the lack of high-quality evidence for the diagnosis of ligament and tendon injuries in the hand and fingers has been highlighted in the current systematic overview of the literature.

Previous reviews showed that a high-resolution MRI was an accurate means for diagnosing TFCC tears and an MRI was slightly specific for tears of the intrinsic ligament, but its sensitivity is low.[10 14] Current review showed that the accuracy measures for an MRI showed a wide

**Table 2** Summary of methodological quality according to Quality Assessment of Diagnostic Accuracy Studies 2

| Author (year), index test(s) | Risk of bias | | | | Applicability concerns | | |
|---|---|---|---|---|---|---|---|
| | Patient selection | Index test | Reference standard | Flow and timing | Patient selection | Index test | Reference standard |
| Wrist disabilities | | | | | | | |
| Anderson et al (2008)[23] | LR | LR | HR | LR | LR | LR | LR |
| Pahwa et al (2014)[32] | UR | LR | HR | HR | LR | LR | LR |
| Prosser et al (2011), provocative tests[33] | LR | LR | LR | LR | LR | LR | LR |
| MRI | LR | LR | HR | LR | LR | LR | LR |
| Langner et al (2015)[40] | LR | LR | HR | HR | LR | LR | LR |
| Spaans et al (2013)[41] | UR | LR | LR | UR | LR | LR | LR |
| Greditzer et al (2016)[24] | HR | LR | HR | LR | LR | LR | LR |
| Al-Hiari (2013)[34] | LR | LR | HR | LR | LR | LR | LR |
| Schmauss et al (2016)[25] | LR | HR | HR | LR | LR | LR | LR |
| Lee et al (2016)[35] | LR | HR | HR | LR | LR | LR | LR |
| Finlay et al (2004)[26] | UR | LR | LR | LR | LR | LR | LR |
| Dornberger et al (2015)[36] | LR | LR | HR | LR | LR | LR | LR |
| Koskinen et al (2012)[42] | LR | HR | HR | LR | LR | LR | LR |
| Boer et al (2018)[27] | HR | UR | HR | LR | LR | LR | LR |
| Lee and Yun (2018)[31] | LR | LR | LR | LR | LR | LR | LR |
| Suojärvi et al (2017)[37] | LR | LR | HR | HR | LR | LR | LR |
| Mahmood et al (2012)[30] | UR | LR | UR | LR | LR | LR | LR |
| Hand and finger disabilities | | | | | | | |
| Lutsky et al (2014)[28] | LR | UR | HR | LR | LR | LR | LR |
| Guntern et al (2007)[29] | LR | HR | LR | LR | LR | LR | LR |
| Klauser et al (2002)[43] | LR | HR | LR | HR | LR | LR | LR |
| Lee et al (2000)[44] | UR | LR | LR | LR | LR | LR | LR |
| Zhang et al (2012)[45] | LR | LR | HR | LR | LR | LR | LR |
| Mahajan et al (2016)[39] | LR | LR | LR | LR | LR | LR | LR |
| Shekarchi et al(2017)[38] | UR | LR | LR | LR | LR | LR | LR |

HR, high risk; LR, low risk; U, unclear risk.

range in diagnostic outcome values, with diagnostic accuracy measures no better for a high-resolution MRI. The present results indicate that the accuracy for tears of the TFCC, SLIL and LTIL is increased by magnetic resonance arthrography (MRA).

### Diagnostic accuracy of the diagnostic tests of the wrist

Although a common practice in hospital care, in previous reviews[10 13–15] and in current systematic review update, no studies were identified on the diagnostic accuracy of history taking for non-chronic ligament and tendon injuries of the wrist.

This systematic review update included one new study on physical examinations for diagnosing non-chronic ligament and tendon injuries of the wrist, which did not affect the previous conclusion that physical examination is of limited value for diagnosing non-chronic ligament and tendon injuries of the wrist.[25]

In previous reviews, only the diagnostic performance for MRI and/or MRA of the wrist were examined. This showed that the accuracy of MRI diagnoses of tears of the TFCC was fairly satisfactory (PPV ranged from 71% to 100% and NPV ranged from 37% to 90% for TFCC, PPV ranged from 25% to 100% and NPV ranged from 72% to 94% for SL ligament and PPV ranged from 0% to 100% and NPV ranged from 74% to 95% for LT ligament) and the best with high-resolution techniques. Contrary, Se, Sp and accuracy were low for diagnosing intrinsic carpal ligaments injuries (SL and LT), using high-resolution techniques.[10 14] MRA, rather than MRI, was recommended to be used in daily practise for the diagnosis of TFCC injuries.[10 15] In the current review, the accuracy measures for an MRI showed a wide range in diagnostic outcome values, with diagnostic accuracy measures no better for imaging at 3 T than at 1.5 T. As previously shown for full-thickness TFCC injuries, the present results indicate that

**Table 3** Accuracy of the diagnostic tests of the wrist

| Author (year) | Index test 1 | Reference test | Trauma | Se (%) (95% CI) | Sp (%) (95% CI) | Accuracy (%) (95% CI) | PPV (%) (95% CI) | NPV (%) (95% CI) |
|---|---|---|---|---|---|---|---|---|
| **Physical examination** | | | | | | | | |
| Prosser et al[33] (2011) | Provocative tests | Arthroscopy | TFCC | 58 | 69 | 73 | 71 | 55 |
| | | | SLIL | 61 | 79 | 78 | 68 | 74 |
| | | | LTIL | 17 | 84 | 95 | 6 | 94 |
| Schmauss et al[25] (2016) | Fovea sign | Arthroscopy | TFCC | 73 | 44 | 58 | 53 | 66 |
| | Ulna grinding test | | | 90 | 20 | 56 | 54 | 65 |
| **Imaging: radiographs** | | | | | | | | |
| Dornberger et al[36] (2015) | Radiographs (Stecher's projection) | Arthroscopy | SLIL | (76.9+80.8)/2* | (86.4+84.1)/2* | (92.7+90.6)/2* | (76.9+75)/2* | (86.4+88.1)/2* |
| **Imaging: US** | | | | | | | | |
| Finlay et al[26] (2004) | US (9–13 MHz) | MR arthrography tricompartment | SLIL | 100 | 100 | 100 | 100 | 100 |
| | | | TFCC | 64 | 100 | 85 | 100 | 79 |
| | | | LTIL | 25 | 100 | 77 | 100 | 75 |
| Lee and Yun[31] (2018) | US | MRI | TFCC, total | 99* | 88* | 97* | 97* | 95* |
| **Imaging: MRI** | | | | | | | | |
| Anderson et al[23] (2008) | MRI (1.5 T) | Arthroscopy | TFCC | 82 | 59 | 83 (72.4 to 89.9)† | | |
| | | | SLIL | 57 | 83 | 78 (67.2 to 86.3)† | | |
| | | | UTIL | 57 | 89 | 95 (86.1 to 98.3)† | | |
| | | | LTIL | 22 | 94 | 86 (75.3 to 91.9)† | | |
| | MRI (3 T) | | TFCC | 90 | 74 | 91 (75.8 to 96.8)† | | |
| | | | SLIL | 70 | 94 | 91 (75.8 to 96.8)† | | |
| | | | UTIL | 67 | 87 | 100 (97.9 to 100)† | | |
| | | | LTIL | 50 | 94 | 91 (75.8 to 96.8)† | | |
| Pahwa et al[32] (2014) | MR arthrography | Arthroscopy | TFCC | 100 | 100 | 100 | 100 | 100 |
| | | | SLIL | 100 | 100 | 100 | 100 | 100 |
| | | | LTIL | 100 | 100 | 100 | 100 | 100 |
| | MRI (1.5 T) MEDIC | | TFCC | 83 | 100 | 81 | 91 | 60 |
| | | | SLIL | 63 | 100 | 81 | 100 | 73 |
| | | | LTIL | 40 | 100 | 81 | 100 | 73 |
| | MRI FS PD/T2 | | TFCC | 75 | 100 | 75 | 90 | 50 |
| | | | SLIL | 38 | 100 | 69 | 100 | 62 |
| | | | LTIL | 20 | 100 | 75 | 100 | 73 |

Continued

**Table 3** Continued

| Author (year) | Index test 1 | Reference test | Trauma | Se (%) (95% CI) | Sp (%) (95% CI) | Accuracy (%) (95% CI) | PPV (%) (95% CI) | NPV (%) (95% CI) |
|---|---|---|---|---|---|---|---|---|
| Prosser et al[33] (2011) | MRI (1 T) | Arthroscopy | TFCC | | | 86 (PT+MRI) | | |
| | | | SLIL | | | 80 (PT+MRI) | | |
| | | | LTIL | | | 94 (PT+MRI) | | |
| Schmauss et al[25] (2016) | MRI resolution not described | Arthroscopy | | 76 | 41 | 58 | 55 | 65 |
| Langner et al[40] (2015) | Cine MRI (3.0 T) and cineradiography | Arthroscopy | SL dissociation | 85 | 90 | 92 | | |
| Spaans et al[41] (2013) | MRI (3 T) | Arthrotomy | SLIL | 75.5* | 100† | 75* | 98.5* | 8† |
| Greditzer et al[24] (2016) | MRI (1.5 T) axial sequences | Arthroscopy | SLIL | 79 | 82 | 80 | 76 | 84 |
| | MRI (1.5 T) coronal sequences | Arthroscopy | SLIL | 65 | 69 | 67 | 68 | 71 |
| Al-Hiari[34] (2013) | MR arthrography | Arthroscopy | TFCC | 93 | 80 | 85 | 87 | 76 |
| Lee[35] (2016) | MR arthrography without traction | Conventional arthrography | TFCC | 83 | 81 | 83 | 87 | 76 |
| | | | SLIL | 66 | 97 | 95 | 67 | 97 |
| | | | LTIL | 57 | 94 | 88 | 67 | 91 |
| | MR arthrography with traction | | TFCC | 96 | 100 | 98 | 100 | 94 |
| | | | SLIL | 100 | 100 | 100 | 100 | 100 |
| | | | LTIL | 100 | 100 | 100 | 100 | 100 |
| Boer et al[27] (2018) | MRI (1.5 T) | Arthroscopy | TFCC | 71 | 75 | 100 | 71 | 75 |
| | MRI (3.0 T) | Arthroscopy | TFCC | 73 | 67 | 89 | 83 | 52 |
| | MR arthrography (1.5 T) | Arthroscopy | TFCC | 80 | 100 | 80 | 100 | 50 |
| | MR arthrography (3.0 T) | Arthroscopy | TFCC | 73 | 100 | 73 | 100 | 60 |
| Suojärvi et al[37] (2017) | MR arthrography | Arthroscopy | SLIL | 25 (3 to 65) | 80 (61 to 92) | 68 (51 to 83) | 25 (3 to 65) | 80 (61 to 92) |
| | | | LTIL | 50 (7 to 93) | 77 (59 to 90) | 74 (57 to 88) | 22 (3 to 60) | 92 (75 to 99) |
| | | | TFCC | 44 (22 to 69) | 50 (25 to 75) | 47 (30 to 65) | 50 (25 to 75) | 44 (21 to 69) |
| | | | SLIL or LTIL | 33 (7 to 60) | 79 (67 to 88) | 72 (56 to 82) | 24 (7 to 50) | 86 (74 to 94) |
| Mahmood et al[30] (2012) | MR arthrography | Arthroscopy | SLIL | 91 | 88 | 88 | 83 | 88 |
| | | | LTIL | 100 | 100 | 100 | 100 | 100 |
| | | | TFCC | 90 | 75 | 73 | 85 | 80 |

Imaging: CT

Continued

**Table 3** Continued

| Author (year) | Index test 1 | Reference test | Trauma | Se (%) (95% CI) | Sp (%) (95% CI) | Accuracy (%) (95% CI) | PPV (%) (95% CI) | NPV (%) (95% CI) |
|---|---|---|---|---|---|---|---|---|
| Koskinen et al[42] (2012) | CBCT arthrography | MR arthrography | TFCC | 76 | 90 | 87 | 83 | 87 |
| | | | SLIL | 56 | 91 | 83 | 67 | 89 |
| | | | LTIL | 83 | 81 | 82 | 44 | 96 |
| Suojärvi et al[37] (2017) | CBCT | Arthroscopy | SLIL | 63 (24 to 91) | 87 (69 to 96) | 82 (66 to 92) | 56 921 to 86 | 90 (73 to 98) |
| | | | LTIL | 100 (40 to 100) | 59 (41 to 76) | 64 (46 to 79) | 24 (7 to 50) | 100 (83 to 100) |
| | | | TFCC | 67 (40 to 87) | 89 (63 to 98) | 77 (60 to 90) | 86 (57 to 98) | 73 (50 to 89) |
| | | | SLIL or LTIL | 75 (43 to 95) | 76 (65 to 86) | 73 (61 to 83) | 35 (16 to 53) | 95 (86 to 99) |

*Average between presented individual values of two readers.
†Only reported for one of two readers.
CBCT, cone-beam CT; FS, fat suppressed; LTIL, lunotriquetral interosseous ligament; MEDIC, multiple-echo data image combination; MR, magnetic resonance; n/a, not available due to low prevalence; NPV, negative predictive value; PD/T2, proton density/tesla2; PPV, positive predictive value; PWT, provocative wrist tests; Se, sensitivity; SLIL, scapholunate interosseous ligament; Sp, specificity; TFCC, triangular fibrocartilage complex; US, ultrasonography; UTIL, ulnotriquetral interosseous ligament.

that the accuracy for tears of the TFCC, SLIL and LTIL is increased by MRA.[15] CT arthrography is an alternative in patients when an MRI is contraindicated or when an MRI is not available.[42]

In the current review, five studies used another imaging tool, namely, radiograph,[36] US[26 31] and CBCT[37 42] for diagnosing non-chronic ligament and tendon injuries of the wrist. The diagnostic accuracy of radiograph was limited. Examination of SLIL and TFCC with US showed promising results and the added value should be further explored. Based on the included studies, CBCT has no added value in assessing non-chronic ligament and tendon injuries of the wrist, especially when we take the methodological quality of the studies into account.

However, a dynamic four-dimensional CT for the detection of SLIL or LTIL injuries is promising.[46 47] Nevertheless, the diagnostic accuracy has not yet been studied. At present, there is still insufficient scientific evidence regarding the ideal imaging technique for non-chronic intrinsic carpal ligament injuries of the wrist.

In the current systematic review update and previous systematic reviews, the reported diagnostic accuracy measures for imaging modalities were characterised by markedly heterogeneous results. It was not appropriate to pool results for the diagnostic accuracy of imaging, due to a lack of multiple imaging studies on one specific wrist injury. Based on previews and the current review, we can conclude that an MRA rather than an MRI is the preferred imaging tool in hospital care setting for detecting non-chronic ligament and tendon injuries of the wrist. The current review focused on diagnostic tests and not on the treatment options for wrist complaints. Arthroscopy, as diagnostic tool, was one of the reference standards in this systematic review. In our opinion, it is essential that readily accessible and relatively inexpensive, non-invasive diagnostics are available to and are preferred by clinicians. For some wrist complaints, arthroscopy may be the preferred diagnostic option. However, it is more expensive and invasive than an MRI. For that reason, diagnostic arthroscopy should be applied with caution, unless a patient is suspected of having non-chronic hand, finger or wrist injury and require therapeutic intervention. The advantage of arthroscopy above MRI is the dynamic modality.

### Diagnostic accuracy of the diagnostic tests of the hand and the fingers

According to our knowledge, there are no reviews previously published to date on the diagnostic accuracy of history taking, physical examination and imaging for non-chronic ligament and tendon injuries of the finger and hand.

We identified three studies on the diagnostic accuracy of history taking and/or clinical examination.[29 39 45] One study[39] had no methodological limitation, while the other two studies had methodological flaws (high risk of bias) on index test[29] and reference standard.[45] In addition, each study evaluated different diagnostics tests for

**Table 4** Accuracy of the diagnostic tests of the hand and fingers

| Author (year) | Index test 1 | Reference test | Trauma | Se (%) (95% CI) | Sp (%) (95% CI) | Accuracy (%) (95% CI) | PPV (%) (95% CI) | NPV (%) (95% CI) |
|---|---|---|---|---|---|---|---|---|
| Lutsky et al[28] (2014) | MRI (open,1.5 T or 3 T) | Surgical findings | Collateral ligament tears of the MPJ of the fingers | 64 | ∞ | 64 | 100 | ∞ |
| Guntern et al[29] (2007) | Clinical examination | MRI (3 T) | A2 pulley lesion | 88 | 100 | 88 | 100 | 95 |
| Klauser et al[43] (2002) | US (12 MHz) | MRI (1.5 T) (and surgical findings, n=7) | Finger pulley injuries | 98 | 100 | 99 | 100 | 97 |
| Lee et al[44] (2000) | US (10–5 MHz) | Surgical findings | Flexor tendon injuries | | | 90 | | |
| Zhang et al[45] (2012) | US (10 MHz) | Surgical findings | Flexor tendon injuries | | | 100 | | |
| | History and clinical examination | | | | | 34 | | |
| Mahajan et al[39] (2016) | Clinical examination | MRI (1.5 T) | UCL injuries | 91 | 75 | 87 | 91 | 75 |
| Shekarchi et al[38] (2017) | US | MRI | UCL of the thumb | 71 (30 to 95) | 85 (54 to 97) | 80 | 71 (30 to 95) | 85 (54 to 97) |

MPJ, metacarpophalangeal joint; NPV, negative predictive value; PPV, positive predictive value; Se, sensitivity; Sp, specificity; UCL, ulnar collateral ligament; US, ultrasonography.

different pathologies. So there is limited evidence on the diagnostic accuracy of history taking and physical examination for diagnosing hand and finger injuries.

Imaging studies examined a wide variety of imaging tools and pathologies. Moreover, studies with imaging tools as a diagnostic modality had methodological flaws and serious limitations, so we have to interpret these results with caution. Only the study of Lee et al had relatively few methodological flaws.[44] These authors showed that US can possibly help to evaluate completely lacerated flexor tendon injuries. Nevertheless, as indicated by the authors, US cannot accurately determine the status of partially transected tendons.[44] The reported diagnostic accuracy measures for imaging modalities were characterised by markedly heterogeneous results. It was not appropriate to pool results for the diagnostic accuracy of imaging, due to the limited number of studies on one specific hand or finger injury and because of the diversity among the eligible studies.

### Clinical care setting

The secondary aim of this study was to include the clinical care setting (hospital or non-institutionalised GP) of the eligible studies and the studies published in previous systematic reviews. We assume that all studies included in the current and previous reviews were done in a hospital care setting; this was either described in the paper, was confirmed by the authors or due to the fact that all authors of the remaining studies were only affiliated to hospitals.

It is essential to distinguish between diagnosing these injuries in hospital care and in non-institutionalised GP care, as results from diagnostic studies in hospital care cannot automatically be translated into guidelines for non-institutionalised GP care.[11] Since previous systematic reviews and the current update of the literature did not identify any studies performed in non-institutionalised GP care, it is not possible to advise GPs with certainty based on the available evidence. Given the burden of non-chronic hand and wrist trauma in non-institutionalised GP care, diagnostic studies focusing on non-chronic hand, finger and wrist ligament and tendon injuries are urgently needed.[1 2]

### CONCLUSIONS

Our systematic review showed that there is still a gap in knowledge regarding valid diagnostic tests for non-chronic wrist ligament and tendon injuries. For the first time, the lack of high-quality evidence for the diagnosis of ligament and tendon injuries in the hand and fingers has been highlighted. Although some imaging modalities seemed to be acceptable for the diagnosis of ligament and tendon injuries in the wrist in patients presenting to secondary care, there are limited tools for adequate diagnosis available to GPs. If not diagnosed and treated properly, patients may experience lifelong pain and functional limitations that have major effects on the quality of life and could result in patients losing their jobs.

**Acknowledgements** The authors thank Wichor M Bramer (biomedical information specialist of Erasmus University Medical Center Rotterdam Medical Library) for help with the electronic search strategies and Yassine Aaboubout (MSc) for helping with study selection and extracting the data.

**Contributors** PK, NMCM, SMAB-Z, GAK and JR all contributed to the design of the study. PK and JR were responsible for article selection and analysed the data. All

authors contributed to writing and revision of the manuscript. All authors have given approval of the submitted version of the manuscript and agree to be accountable for all aspects of the work.

**Funding** The authors have not declared a specific grant for this research from any funding agency in the public, commercial or not-for-profit sectors.

**Competing interests** None declared.

**Patient and public involvement** Patients and/or the public were not involved in the design, or conduct, or reporting, or dissemination plans of this research.

**Patient consent for publication** Not required.

**Provenance and peer review** Not commissioned; externally peer reviewed.

**Data availability statement** Data are available upon reasonable request. The datasets used and/or analysed during the current study are available from the corresponding author on reasonable request.

**ORCID iD**
Patrick Krastman http://orcid.org/0000-0002-5609-9471

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
