## [Reviewer comments · BMJ Open]

ARTICLE DETAILS

TITLE (PROVISIONAL)	Diagnostic accuracy of history taking, physical examination and imaging for non-chronic finger, hand and wrist ligament and tendon injuries: a systematic review update.
AUTHORS	Krastman, Patrick; Mathijssen, Nina; Bierma-Zeinstra, Sita; Kraan, Gerald; Runhaar, Jos

VERSION 1 - REVIEW

REVIEWER	Jonny K Andersson Aspetar Orthopaedic and Sports Medicine Hospital, Doha, Qatar and The Sahlgrenska Academy, Department of Orthopaedics, Institute of Clinical Sciences, University of Gothenburg, Göteborg, Sweden
REVIEW RETURNED	02-May-2020

GENERAL COMMENTS	Thank you for letting me review this interesting and well written manuscript. I think that there are quite many authors – can they explain how they have contributed to the study? Accuracy is a difficult term to understand (at least in my opinion) – can the authors explain what Accuracy is and how you calculate that? When calculating Sensitivity, Specificity, PPV and NPV, you always have to have a "gold standard". What was the gold standard in this study – findings during open surgery, arthroscopy? Seems to be different in the different selected studies after search strategies, as diagnostic-imaging techniques for hand injuries seems to be accepted. In my opinion you then compare apples with pears - Comments!?. I would prefer if the authors could discuss their findings and compare their results with the findings in the following studies more in their Discussion section: Andersson JK, Andernord D, Karlsson J, Fridén J. Efficacy of magnetic resonance imaging and clinical tests in diagnostics of wrist ligament injuries: a systematic review. Arthroscopy. 2015, 31: 2014-20. And Hobby JL, Tom BD, Bearcroft PW, Dixon AK. Magnetic resonance imaging of the wrist: Diagnostic performance statistics. Clin Radiol 2001;56:50-57. Abstract: Page 2, Line 38 and 45: Please define what you mean by Accuracy and how Accuracy really is calculated. What are your thoughts about
---

this often used term Accuracy?
Page 3, Line 14 – Is this really the first study that systematically reviews diagnostic test of hand, finger and wrist injuries? Please check references.

Introduction:

Page 6, Line 6: I understand that hand surgeons performing tests for finger, hand and wrist injuries are more dedicated and (maybe) have a higher sensitivity and specificity in finding different clinical signs of different injuries in the clinical setting - but there should in my opinion be no difference in the accuracy of diagnostics by MRI. Therefore I do not understand the table 1, where both Physical examination and Imaging are shown. Do you really think there is a difference in imaging reports, if the referrals come from GPs or hand surgeons? Could be! In that case, I recommend that you refer to: Ringler MD. MRI of wrist ligaments. J Hand Surg Am. 2013, 38: 2034-46. He underlines the importance of a dedicated radiologist and the close cooperation between the clinician and the radiologist in diagnostics of wrist ligament injuries.

Methods:

What were your Primary measure outcome and Secondary measure outcomes? Please define more accurate. A systematic should always include that, otherwise it is more like a literature review.

Page 7, Line 47: Why was distal radius and ulna fractures excluded. Approximately 40 % of all dislocated distal radius fractures have concomitant wrist ligament injuries. Possible risk of inclusion bias.

Page 7, Line 56: See above mention question about gold standard.

Page 8, line 54: Good that you have used the QUADAS-2 tool!

Page 9, line 15: If the articles were written in another language than English – who did perform the translations?

Results:

The tables under this section are huge with big empty spaces. Could be done much better. Difficult to read and get a proper overview.

Discussion:

Start this section with your most important findings - Not your purpose – belongs to introduction.

As you mention Andersson JK, Andernord D, Karlsson J, Fridén J. Efficacy of magnetic resonance imaging and clinical tests in diagnostics of wrist ligament injuries: a systematic review. Arthroscopy. 2015, 31: 2014-20.

And

Hobby JL, Tom BD, Bearcroft PW, Dixon AK. Magnetic resonance imaging of the wrist: Diagnostic performance statistics. Clin Radiol 2001;56:50-57, in the introduction, you should comment on those articles more in your discussion.

Exactly what more information do you believe that you have been able to extract in your current study, compared to those former systematic reviews?

I also miss a clinical comment about that wrist arthroscopy is both diagnostic and therapeutic and the only tool that can evaluate grade of instability and healing capacity. Waiting too long for an MRI instead of going for a rapid direct re-insertion of the SL ligament (should be performed in 4-6 w), could also be hazardous...

References are updated and the Figure of search strategies is good.

	This is a well written manuscripts. I suggest minor revision and I am looking forward to answers and explanations to my above comments. Jonny K Andersson MD, PhD
--	--

REVIEWER	Kenneth J Young University of Central Lancashire, UK
REVIEW RETURNED	04-May-2020

GENERAL COMMENTS	This systematic review addresses a worthy topic. It is had a defined research question that is adequately addressed. It is generally well-written and uses appropriate references. The tables could be reformatted for easier reading, and a few sentences or word uses are awkward. Please see attached document for specific comments on the text. The reviewer provided a marked copy with additional comments. Please contact the publisher fo full details.
--

REVIEWER	Nima H Nejad Johns Hopkins University
REVIEW RETURNED	03-Jun-2020

GENERAL COMMENTS	I reviewed the manuscript entitled “Diagnostic accuracy of history taking, physical examination and imaging for non-chronic finger, hand and wrist ligament and tendon injuries: a systematic review update.” I have a few recommendations to improve the study presentation. 1. The authors have undertaken a big task trying to review the diagnostic accuracy of all pertinent information from history, physical examination, and diagnostic imaging examinations. One of the cornerstones of a well carved out systematic review is the homogeneity of the study objectives and included manuscripts. Because of the broad scope of this review and heterogeneous nature of the included studies, no meta-analysis was performed. It may be worthwhile to emphasize heterogeneity as one of the main drawbacks of this study. There is remarkable heterogeneity for diagnostic evaluations, outcomes, and natural history of the studied entities (i.e. “finger, hand, and wrist ligament injuries” covers a wide variety of pathologies that may require different sets of evaluations and be associated with different outcomes). Even in terms of diagnostic imaging, workups, techniques, and protocols can vary significantly depending on pathologies. 2. Conclusions: The authors mentioned: “there is limited evidence on which imaging tool is most accurate for diagnosing non-chronic ligament and tendon injuries of the hand and/or fingers.”. In terms of diagnostic accuracy, most ligaments are best visualized using high-resolution MRI (again, it’s very hard to put all pathologies and conditions under a single umbrella of “non-chronic wrist, hand or finger pain”). MRA may be more helpful when visualization of an articular structure is desired. The better way to phrase the conclusion may be: there is limited evidence on which imaging tool is more “feasible” or “cost-effective” rather than “accurate”.
--

	3. Tables: Given the extent of the provided information, having well summarized and organized tables is of critical importance. For example, table 1 may benefit from summarization (if there is not enough space on a portrait orientation, add the table to the end of the manuscript or a separate file and use the landscape orientation). If the provided numbers for diagnostic accuracy are actual percentages, please consider re-writing them accordingly (i.e. 80% instead of 0.80 when labeled with %). It is uncommon to include a table in "introduction". Since this is a systematic review and table 1 is meant to summarize some of the prior studies, please consider moving it to results. 4. Appendix 1: Was EMBASE the only database that was searched? (it appears that MEDLINE and Web of Science were also evaluated). Please consider describing the search criteria for other databases. 5. Please consider defining and describing the exclusion criteria. 6. Flow chart: Excluded studies: Please elaborate more on the "Age population (58)" (not sure what "age population" means here). "Systemic/Chronic (12)" Why were chronic cases excluded here? Please clarify. 7. Table 4 is 4 pages long and would benefit from being better summarized. Authors may consider having a more organized presentation of their tables rather than including as much information as possible.
--	--

VERSION 1 – AUTHOR RESPONSE

Reviewer 1 (Jonny K Andersson)

Comment reviewer 1:

I think that there are quite many authors – can they explain how they have contributed to the study?

Authors' reply:

All authors comply with the requirements for authorship, based on the International Committee of Medical Journal Editors (ICMJE).

As depicted on page 25 of our revised manuscript (clean copy): "PK, NM, SB, GK and JR all contributed to the design of the study. PK and JR were responsible for article selection and analysed the data. All authors contributed to writing and revision of the manuscript. All authors have given approval of the submitted version of the manuscript and agree to be accountable for all aspects of the work".

Authors' action:

No action taken.

Comment reviewer 1:

Can the authors explain what Accuracy is, how you calculate that and what are your thoughts about this often used term Accuracy?

Authors' reply:

Diagnostic accuracy relates to the ability of a test to discriminate between the presence or absence of an injury. Different measures of diagnostic accuracy relate to the different aspects of diagnostic procedure: while some measures are used to assess the discriminative property of the test, others are used to assess its predictive ability. [Ana-Maria Šimundić Measures of Diagnostic Accuracy: Basic Definitions. EJIFCC. 2009 Jan; 19(4): 203–211.][Paolo Eusebi. Diagnostic Accuracy Measures. Cerebrovasc Dis Actions. 2013;36(4):267-72.]

The following formula was used, when calculating diagnostic accuracy:

Diagnostic accuracy = No. of true positives + no. of true negatives / Total no. of subjects

[Worster A. and Carpenter C. Incorporation bias in studies of diagnostic tests: how to avoid being biased about bias. CJEM 2008;10(2):174-5]

Within the focus of the current manuscript, diagnostic accuracy is the ability of a diagnostic test to distinguish between "the presence of hand or wrist pathology" and "the absence of hand or wrist pathology." This makes it an essential part of the evaluation of a diagnostic test and has to be complete and transparent according to the STARD statement. [STARD 2015: an updated list of essential items for reporting diagnostic accuracy studies. BMJ. 2015; 351: h5527. Published online 2015 Oct 28. Patrick M Bossuyt,¹ Johannes B Reitsma,² David E Bruns,³ Constantine A Gatsonis,⁴ Paul P Glasziou,⁵ Les Irwig,⁶ Jeroen G Lijmer,⁷ David Moher,^{8,9} Drummond Rennie,^{10,11} Henrica C W de Vet,¹² Herbert Y Kressel,^{13,14} Nader Rifai,^{15,16} Robert M Golub,^{17,18} Douglas G Altman,¹⁹ Lotty Hooft,²⁰ Daniël A Korevaar,¹ and Jérémie F Cohen^{1,21}, for The STARD Group]

Authors' action:

The formula, calculating diagnostic accuracy, is implemented on page 7, line 15 of our revised manuscript (clean copy).

Comment reviewer 1:

What was the gold standard in this study – findings during open surgery, arthroscopy?

Authors' reply:

Unfortunately, in a diagnostic process of finger, hand and wrist ligament and tendon pathologies, there is no diagnostic gold standard. For that reason, as described in the revised manuscript (page 6, lines 10-15) all forms of reference standard were accepted in current review and no articles were excluded based on the used reference standard.

Authors' action:

We add your remark in the section methods (page 6, line 9): "There was no gold-standard reference test against which to assess history taking, physical examination or imaging measurements.

Comment reviewer 1:

The reference standard seems to be different in the different selected studies after search strategies, as diagnostic-imaging techniques for hand injuries seems to be accepted. In my opinion you then compare apples with pears - Comments?

Authors' reply:

In our systematic review, we wanted to be as complete as possible and not exclude studies, based on a different reference standard (surgical findings and MRI). Unfortunately, in the current review a meta-analysis was not possible, due to the limited number of studies on one specific hand or finger injury and because of the diversity among the eligible studies (e.g. pathology, index test, reference standard and QUADAS).

Authors' action:

In response to the reviewer's question, we decided to add the following sentence to what has been described above, page 22, line 8. "The reported diagnostic accuracy measures for imaging modalities were characterized by markedly heterogeneous results. It was not appropriate to pool results for the diagnostic accuracy of imaging, due to the limited number of studies on one specific hand or finger injury and because of the diversity among the eligible studies.

Comment reviewer 1:

I would prefer if the authors could discuss their findings and compare their results with the findings in the following studies more in their Discussion section. [Andersson JK, Andersnord D, Karlsson J, Fridén J. Efficacy of magnetic resonance imaging and clinical tests in diagnostics of wrist ligament injuries: a systematic review. Arthroscopy. 2015, 31: 2014-20.][Hobby JL, Tom BD, Bearcroft PW, Dixon AK. Magnetic resonance imaging of the wrist: Diagnostic performance statistics. Clin Radiol 2001;56:50-57.]

Authors' reply:

Thank you for this valuable comment. As mentioned by the reviewer, both are important reviews. We summarize their findings in the following way:

In previous reviews, only the diagnostic performance for MRI and/or MR Arthrography (MRA) of the wrist were examined. This showed that the accuracy of MRI diagnoses of tears of the TFCC was fairly satisfactory (PPV ranged from 71% to 100% and NPV ranged from 37% to 90% for TFCC, PPV ranged from 25% to 100% and NPV ranged from 72% to 94% for SL ligament, and PPV ranged from 0% to 100% and NPV ranged from 74% to 95% for LT ligament) and the best with high-resolution techniques. Contrary, sensitivity, specificity, and accuracy were low for diagnosing intrinsic carpal ligaments injuries (SL and LT), using high-resolution techniques.

Authors' action:

In the discussion section (page 20, line 1) of our revised manuscript (clean copy), we have replaced the previous quote: "In previous reviews, only the diagnostic performance for MRI and/or MR Arthrography (MRA) of the wrist were examined. This showed that high-resolution MRI was an accurate means for diagnosing TFCC tears and MRI was slightly specific for tears of the intrinsic ligament, but its sensitivity is low." with the new quote (see above).

Comment reviewer 1:

Page 3, Line 14 – Is this really the first study that systematically reviews diagnostic test of hand, finger and wrist injuries? Please check references.

Authors' reply:

Your statement is partly correct. This is the first study that systematically reviewed the accuracy of diagnostic tests for non-chronic hand and finger injuries. As described in current review, this is not the first study that systematically reviewed the accuracy of diagnostic tests for non-chronic wrist injuries.

Authors' action:

No action taken

Comment reviewer 1:

Introduction: Page 6, Line 6: I understand that hand surgeons performing tests for finger, hand and wrist injuries are more dedicated and (maybe) have a higher sensitivity and specificity in finding different clinical signs of different injuries in the clinical setting - but there should in my opinion be no difference in the accuracy of diagnostics by MRI. Therefore I do not understand the table 1, where both Physical examination and Imaging are shown. Do you really think there is a difference in imaging reports, if the referrals come from GPs or hand surgeons? Could be! In that case, I recommend that you refer to: Ringler MD. MRI of wrist ligaments. J Hand Surg Am. 2013, 38: 2034-46. He underlines the importance of a dedicated radiologist and the close cooperation between the clinician and the radiologist in diagnostics of wrist ligament injuries.

Authors' reply:

Diagnostic accuracy is affected by the prevalence of the pathology. With the same sensitivity and specificity, diagnostic accuracy of a particular test increases as the prevalence of the pathology increases. Unlike sensitivity and specificity, predictive values are largely dependent on the prevalence of the pathology in the examined population. Therefore, predictive values from one study should not be transferred to another setting with a different prevalence of the disease in the population.

Prevalence affects PPV and NPV differently. PPV is increasing, while NPV decreases with the increase of the prevalence of the disease in a population. Whereas the change in PPV is more substantial, NPV is somewhat weaker influenced by the disease prevalence. [Ana-Maria Šimundić Measures of Diagnostic Accuracy: Basic Definitions. EJIFCC. 2009 Jan; 19(4): 203–211.]

Authors' action:

In the introduction section (page 4, line 25) of our revised manuscript (clean copy), we have implemented the following sentence: "Diagnostic accuracy is affected by the prevalence of the pathology. Predictive values are largely dependent on the prevalence of the pathology in the examined population. Therefore, predictive values from one study should not be transferred to

another setting with a different prevalence of the disease in the population. [Ana-Maria Šimundić Measures of Diagnostic Accuracy: Basic Definitions. EJIFCC. 2009 Jan; 19(4): 203–211.]”.

Comment reviewer 1:

Methods: What were your Primary measure outcome and Secondary measure outcomes? Please define more accurately. A systematic should always include that, otherwise it is more like a literature review.

Authors' reply:

In the current review our primary outcome measures were the positive predictive value (PPV) and the negative predictive value (NPV) of diagnostic tests. Secondary outcome measures were the sensitivity, specificity and accuracy of diagnostic tests.

Authors' action:

In the method section (page 7, line 5) of our revised manuscript (clean copy): “Data collection process and Methodological Quality Assessment”, we insert the primary and secondary outcome measurement descriptions.

Comment reviewer 1:

Page 7, Line 47: Why were distal radius and ulna fractures excluded. Approximately 40 % of all dislocated distal radius fractures have concomitant wrist ligament injuries. Possible risk of inclusion bias.

Authors' reply:

To us, diagnostic accuracy of tests for distal radius and ulna fractures is another and interesting topic. We agree with the reviewer that distal radius and ulna fractures often have concomitant wrist ligament injuries. However, the current review aimed to evaluate the diagnostic accuracy of tests specifically for non-chronic finger, hand and wrist ligament and tendon injuries and not the suggested fractures. For a review on fracture diagnostic testing, see our recent review: “Kraştman P, Mathijssen NM, Bierma-Zeinstra SMA, Kraan G, Runhaar J. Diagnostic accuracy of history taking, physical examination and imaging for phalangeal, metacarpal and carpal fractures: a systematic review update. BMC Musculoskelet Disord. 2020 Jan 7;21(1):12.”.

Authors' action:

No action taken

Comment reviewer 1:

Page 7, Line 56: See above mention question about gold standard.

Authors' reply:

See previous reply

Authors' action:

No action taken

Comment reviewer 1:

Page 8, line 54: Good that you have used the QUADAS-2 tool!

Authors' reply:

Thank you for the positive feedback.

Authors' action:

No action taken

Comment reviewer 1:

Page 9, line 15: If the articles were written in another language than English – who did perform the translations?

Authors' reply:

If the articles were not written in another language than English, Google translate was used for the first translation of the studies. If necessary, a professional translator was consulted. [Accuracy of Data Extraction of Non-English Language Trials with Google Translate. [Balk EM, Chung M, Hadar N, Patel K, Yu WW, Trikalinos TA, Chang LKW. Balk EM, et al. Rockville (MD): Agency for Healthcare Research and Quality (US); 2012 Apr.]

Authors' action:

In the method section (page 6, line 21) of our revised manuscript (clean copy), after the sentence, "No language restrictions were applied", we added the following sentence: "For languages of the eligible studies other than English, Google translate was used for the first translation of these studies. If necessary, a professional translator was consulted. [Accuracy of Data Extraction of Non-English Language Trials with Google Translate. [Balk EM, Chung M, Hadar N, Patel K, Yu WW, Trikalinos TA, Chang LKW. Balk EM, et al. Rockville (MD): Agency for Healthcare Research and Quality (US); 2012 Apr.]"

Comment reviewer 1:

Results: The tables under this section are huge with big empty spaces. Could be done much better. Difficult to read and get a proper overview.

Authors' reply:

Where possible, adjustments to the tables have been made in the revised manuscript. We have also kept the layout as simple as possible.

Authors' action:

See tables in Main document

Comment reviewer 1:

Discussion: Start this section with your most important findings - Not your purpose – belongs to introduction. Exactly what more information do you believe that you have been able to extract in your current study, compared to those former systematic reviews? you should comment on those articles more in your discussion. [Andersson JK, Andernord D, Karlsson J, Fridén J. Efficacy of magnetic resonance imaging and clinical tests in diagnostics of wrist ligament injuries: a systematic review. Arthroscopy. 2015, 31: 2014-20.][Hobby JL, Tom BD, Bearcroft PW, Dixon AK. Magnetic resonance imaging of the wrist: Diagnostic performance statistics. Clin Radiol 2001;56:50-57]

Authors' reply:

Our most important findings are that there still is a gap in knowledge regarding valid diagnostic tests for non-chronic wrist ligament and tendon injuries. Moreover, for the first time, the lack of high-quality evidence for the diagnosis of ligament and tendon injuries in the hand and fingers has been highlighted in the current systematic overview of the literature.

Previous reviews showed that high-resolution MRI was an accurate means for diagnosing TFCC tears and MRI was slightly specific for tears of the intrinsic ligament, but its sensitivity is low. 10 13 Current review showed that the accuracy measures for MRI showed a wide range in diagnostic outcome values, with diagnostic accuracy measures no better for high-resolution MRI. The present results indicate that the accuracy for tears of the TFCC, SLIL and LTIL is increased by MRA.

Authors' action:

In the discussion section (page 20, line 5) of our revised manuscript (clean copy), after the sentence, "There is general agreement that a detailed patient history and a conscientious clinical examination should be standard methods of diagnosing wrist pain.⁹, we added the following sentences: "Our systematic review showed that there still is a gap in knowledge regarding valid diagnostic tests for non-chronic wrist ligament and tendon injuries. Moreover, for the first time, the lack of high-quality evidence for the diagnosis of ligament and tendon injuries in the hand and fingers has been highlighted in the current systematic overview of the literature.

Previous reviews showed that high-resolution MRI was an accurate means for diagnosing TFCC tears and MRI was slightly specific for tears of the intrinsic ligament, but its sensitivity is low. 10 13 Current review showed that the accuracy measures for MRI showed a wide range in diagnostic outcome

values, with diagnostic accuracy measures no better for high-resolution MRI. The present results indicate that the accuracy for tears of the TFCC, SLIL and LTIL is increased by MRA.

Comment reviewer 1:

See discussion, I also miss a clinical comment about that wrist arthroscopy is both diagnostic and therapeutic and the only tool that can evaluate grade of instability and healing capacity. Waiting to long for an MRI instead of going for a rapid direct re-insertion of the SL ligament (should be performed in 4-6 w), could also be hazardous...

Authors' reply:

The current review focused on diagnostic tests and not on the treatment options for wrist complaints. Arthroscopy, as diagnostic tool, was one of the reference standards in this systematic review. In our opinion is it essential that readily accessible and relatively inexpensive, non-invasive diagnostics are available to and are preferred by clinicians. For some wrist complaints, arthroscopy may be the preferred diagnostic option. However, it is more expensive and invasive than MRI. For that reason diagnostic arthroscopy should be applied with caution, unless a patient is suspected of having non-chronic hand, finger or wrist injury and require therapeutic intervention. The advantage of arthroscopy above MRI is the dynamic modality. Furthermore, despite being the gold standard for diagnostic work-up for persistent ulnar-sided wrist pain, wrist arthroscopy is associated with considerable intra- and interobserver variability. [Lów S, Erne H, Schütz A, Eingartner C, Spies CK (2015) The required minimum length of video sequences for obtaining a reliable interobserver diagnosis in wrist arthroscopies. Arch Orthop Trauma Surg 135:1771–1777][Lów S, Pillukat T, Prommersberger KJ, van Schoonhoven J (2013) The effect of additional video documentation to photo documentation in wrist arthroscopies on intra- and interobserver reliability. Arch Orthop Trauma Surg 133:433–438][Lów S, Herold A, Eingartner C (2014) Standard wrist arthroscopy: technique and documentation. Oper Orthop Traumatol 26:539–546].

Authors' action:

In the discussion section (page 22, line 6) of our revised manuscript (clean copy), after the sentence, "Based on previews and current review we can conclude that MRA rather than MRI is the preferred imaging tool in hospital care setting for detecting non-chronic ligament and tendon injuries of the wrist.", we added the following sentences: "The current review focused on diagnostic tests and not on the treatment options for wrist complaints. Arthroscopy, as diagnostic tool, was one of the reference standards in this systematic review. In our opinion is it essential that readily accessible and relatively inexpensive, non-invasive diagnostics are available to and are preferred by clinicians. For some wrist complaints, arthroscopy may be the preferred diagnostic option. However, it is more expensive and invasive than MRI. For that reason diagnostic arthroscopy should be applied with caution, unless a patient is suspected of having non-chronic hand, finger or wrist injury and require therapeutic intervention. The advantage of arthroscopy above MRI is the dynamic modality."

Reviewer: 2 (Kenneth J Young)

Comment reviewer 2:

This systematic review addresses a worthy topic. It is had a defined research question that is adequately addressed. It is generally well-written and uses appropriate references.

Authors' reply:

Thank you for the positive feedback.

Authors' action:

No action required

Comment reviewer 2:

The tables could be reformatted for easier reading, and a few sentences or word uses are awkward.

Authors' reply:

Where possible, adjustments to the tables have been made (e.g. we changed the page for the table to landscape mode). We have also kept the layout as simple as possible.

Authors' action:

See tables of our revised manuscript (clean copy).

Comment reviewer 2:

Table 2, use the same format in the boxes below the heading (i.e. delete the comma after 'Author' and put 'year' in parentheses.)

Authors' reply:

We would like to thank the reviewer for his recommendation.

Authors' action:

See table 2 for the adjustments of our revised manuscript (clean copy).

Comment reviewer 2:

Page 21, line 8: Beside MRA, rather than MRI, was recommended to be used in daily practise for the diagnosis of TFCC injuries.10 14 Reviewer 2 is not sure of the meaning here. Reword?

Authors' reply:

We would like to thank the reviewer for his recommendation.

Authors' action:

Sentence has been rewritten: "MRA, rather than MRI, was recommended to be used in daily practise for the diagnosis of TFCC injuries.10 14"

Comment reviewer 2:

Discussion, page 24, line 28: Replace the word "Beside" with the word "In addition".

Authors' reply:

We would like to thank the reviewer for his recommendation.

Authors' action:

Recommendation has been implemented (page 22, line 22).

Comment reviewer2:

Discussion, page 23, line 1: "Imaging studies exhibit a wide variety of imaging tools and studied pathologies". Replace sentences by "Imaging studies examined a wide variety of imaging tools and pathologies."

Authors' reply:

We would like to thank the reviewer for his recommendation.

Authors' action:

Recommendation has been implemented.

Comment reviewer2:

Discussion, page 23, line 12: "The secondary aim of this study was to retrieve the clinical care setting (hospital or noninstitutionalized GP) of the eligible studies and the studies published in previous systematic reviews. "Replace the word "retrieve" with the word "include".

Authors' reply:

We would like to thank the reviewer for his recommendation.

Authors' action:

Recommendation has been implemented.

Comment reviewer2:

Discussion, page 23, line 19: "Since previous systematic reviews and the current update of the literature did not identify any studies performed in non-institutionalized GP care, it is not possible to advise GPs conscientiously. "Replace the word "conscientiously" with the words "with certainty based on the available evidence".

Authors' reply:

We would like to thank the reviewer for his recommendation.

Authors' action:

Recommendation has been implemented.

Comment reviewer 2:

Conclusion: The first sentence is okay, however the rest are not really conclusions but rather just a summary of results. What action would you recommend from your results? What further research would you suggest? Be specific.

Authors' reply:

We agree with the reviewer that the conclusion could be rewritten.

Authors' action:

Conclusion has been rewritten and implemented (page 24, line 32) of the revised manuscript. "Our systematic review showed that there still is a gap in knowledge regarding valid diagnostic tests for non-chronic wrist ligament and tendon injuries. For the first time, the lack of high-quality evidence for the diagnosis of ligament and tendon injuries in the hand and fingers has been highlighted. Although some imaging modalities seemed to be acceptable for the diagnosis of ligament and tendon injuries in the wrist in patients presenting to secondary care, there are limited tools for adequate diagnosis available to general practitioners. If not diagnosed and treated properly, patients may experience lifelong pain and functional limitations that have major effects on the quality of life and could result in patients losing their jobs."

Reviewer: 3 (Nima H Nejad)

Comment reviewer 3:

It may be worthwhile to emphasize heterogeneity as one of the main drawbacks of this study.

Authors' reply:

We would like to thank the reviewer for this comment. The purpose of the present study was to provide a systematic overview of the diagnostic accuracy of history taking, physical examination and imaging for detecting non-chronic ligament and tendon injuries of the finger, hand and wrist. Trying to review the diagnostic accuracy of all pertinent information from history, physical examination, and diagnostic imaging examinations was not the main cause of heterogeneity. The main cause of heterogeneity in this review was caused by the fact that studies that evaluated the same pathologies showed marked diversity in population, index tests, reference test and methodological quality. In the strengths and limitations of this study, this is discussed: "Diagnostic tests heterogeneity precluded meta-analysis".

We would like to thank the reviewer for his recommendation and we rewritten this sentence.

Authors' action:

In the section strengths and limitation of the study the sentence "Diagnostic tests heterogeneity precluded meta-analysis" has been rewritten: "Diagnostic tests heterogeneity precluded meta-analysis, caused by the fact that studies that evaluated the same pathologies showed marked diversity in population, index tests, reference test and methodological quality."

Comment reviewer 3:

The better way to phrase the conclusion may be: there is limited evidence on which imaging tool is more "feasible" or "cost-effective" rather than "accurate".

Authors' reply:

We would like to thank the reviewer for his recommendation and we rewritten the discussion.

Authors' action:

Conclusion on page 2 has been rewritten.

There is limited evidence on the diagnostic accuracy of history taking and physical examination for non-chronic finger, hand and wrist ligament and tendon injuries. Although some imaging modalities seemed to be acceptable for the diagnosis of ligament and tendon injuries in the wrist in patients

presenting to secondary care, there is no evidence-based advice possible for the diagnosis of non-chronic finger, hand, or wrist ligament and tendon injuries in primary care.

Comment reviewer 3:

Tables: Given the extent of the provided information, having well summarized and organized tables is of critical importance. For example, table 1 may benefit from summarization (if there is not enough space on a portrait orientation, add the table to the end of the manuscript or a separate file and use the landscape orientation).

Authors' reply:

Where possible, adjustments to the tables have been made. We have also kept the layout as simple as possible.

Authors' action:

See tables in Main document

Comment reviewer 3:

If the provided numbers for diagnostic accuracy are actual percentages, please consider re-writing them accordingly (i.e. 80% instead of 0.80 when labelled with %).

Authors' reply:

We would like to thank the reviewer for his recommendation.

Authors' action:

We have decided to remove table 1.

Comment reviewer 3:

It is uncommon to include a table in "introduction". Since this is a systematic review and table 1 is meant to summarize some of the prior studies, please consider moving it to results.

Authors' reply:

We would like to thank the reviewer for his recommendation.

Authors' action:

In reply to the comments of the reviewer and the numbers of the tables in the current review, we have decided to remove table 1. All studies on finger, hand and wrist injuries published before 2000 were adequately discussed in written form in the current review.

Comment reviewer 3:

Appendix 1: Was EMBASE the only database that was searched? (it appears that MEDLINE and Web of Science were also evaluated). Please consider describing the search criteria for other databases.

Authors' reply:

As described on page 7, the search strategy was done in Medline, Embase, Cochrane Library, Web of Science, Google scholar ProQuest and Cinahl. Only as an example, the EMBASE search was provided in the appendix.

Authors' action:

No action taken.

Comment reviewer 3:

Please consider defining and describing the exclusion criteria.

Authors' reply:

We would like to thank the reviewer for his recommendation.

Authors' action:

The selection criteria has been rewritten, it now read (page 6, line 4):

"Studies describing diagnostic accuracy of history taking, physical examination or imaging in adult patients (age \geq 16 years) with non-chronic finger, hand and wrist ligament and tendon injuries were

included. Diagnostic accuracy was reported or could be calculated. Case reports, reviews and conference proceedings were excluded. Distal radius and ulna injuries were also excluded.”

Comment reviewer 3:

Flow chart: Excluded studies: Please elaborate more on the “Age population (58)” (not sure what “age population” means here). Please clarify.

Authors’ reply:

In our selection criteria we described that studies describing diagnostic accuracy of history taking, physical examination or imaging in adult patients (age \geq 16 years) with non-chronic finger, hand and wrist ligament and tendon injuries were included.

Authors’ action:

Flow diagram: “Age population” is replaced by “Age population (age < 16 years)”.

Comment reviewer 3:

“Systemic/Chronic (12)” Why were chronic cases excluded here? Please clarify.

Authors’ reply:

The topic of the systematic overview is non-chronic finger, hand and wrist ligament and tendon injuries. By non-chronic injuries, we mean acute and sub-acute complaints of finger, hand and wrist ligament and tendon injuries. Chronic complaints has a different pathophysiology. In the context of heterogeneity the chronic cases were excluded.

Authors’ action:

After the selection criteria (page 6, line 8) we implemented the following sentence: “Chronic injuries (e.g. osteoarthritis) were excluded as a result of another pathophysiology.”

Comment reviewer 3:

Table 4 is 4 pages long and would benefit from being better summarized. Authors may consider having a more organized presentation of their tables rather than including as much information as possible.

Authors’ reply:

We agree with the comment and where possible, adjustments to the tables have been made. We have also kept the layout as simple as possible.

Authors’ action:

See tables in Main document.

VERSION 2 – REVIEW

REVIEWER	Kenneth J Young University of Central Lancashire, UK
REVIEW RETURNED	09-Jul-2020
GENERAL COMMENTS	I believe that all the comments by the reviewers have been adequately addressed.